# Evaluation of the setup discrepancy between 6D ExacTrac and cone beam computed tomography in spine stereotactic body radiation therapy

**Jaehyeon Park[1,2], Ji Woon Yea[1,2], Jae Won Park[1,2], Se An Oh[1,2]***

1 Department of Radiation Oncology, Yeungnam University Medical Center, Daegu, South Korea,
2 Department of Radiation Oncology, Yeungnam University College of Medicine, Daegu, South Korea

* sean.oh5235@gmail.com

**Data Availability Statement:** All relevant data are within the paper and its Supporting Information files.

## Abstract

The objective of this study was to analyze the difference in residual setup errors between 6D ExacTrac and 3D cone-beam computed tomography (CBCT) image-guided systems in spinal stereotactic body radiation therapy (SBRT). We investigated 76 patients with spinal tumors who received SBRT using Novalis Tx at our institution between January 2013 and September 2020. A Vac-lok (EZ-FIX®, Arlico Medical Company, South Korea) fixture and an assistive device, based on the region involved, were used to immobilize patients and to increase the inter-fractional setup reproducibility. The difference in the root mean square (RMS) between the 6D ExacTrac and 3D CBCT was -0.75 mm, 0.45 mm, 0.16 mm, and -0.03˚; the RMS value was 1.31 mm, 1.06 mm, 0.87 mm, and 0.64˚; and the standard deviation was 0.80 mm, 0.72 mm, 0.62 mm, and 0.44˚ for lateral, longitudinal, vertical, and yaw directions, respectively. The difference in the average RMS between ExacTrac and CBCT was <1.03 mm in the translation direction and <0.47˚ in the rotational direction; the results were statistically significant in the lateral, longitudinal, and vertical directions, but not in the yaw direction. Thus, it is necessary to verify the ExacTrac image according to the CBCT image.

## Introduction

The clinical efficacy of stereotactic body radiation therapy (SBRT) for spinal tumors has been previously reported [1–6]. Precise delivery of high doses of radiation in spinal tumors with SBRT has shown potential clinical benefits. To perform sophisticated treatments such as stereotactic radiosurgery (SRS) and SBRT, the use of ExacTrac (BrainLAB, Feldkirchen, Germany) and cone-beam computed tomography (Varian Medical System, CA, USA) image guidance systems is essential [7–10].

SBRT delivers high doses per fraction; therefore, an extremely steep dose gradient is required to deliver minimum and maximum radiation doses to the normal organ and tumors, respectively [8]. To minimize the radiation dose to the normal organs, a minimal setup margin

**Funding:** This work was supported by the 2020 Yeungnam University Research Grant. The funders had no role in study design, data collection and analysis, decision to publish, or preparation of the manuscript.

**Competing interests:** The authors have declared that no competing interests exist.

is required for the tumor [7]. In contrast, if the setup margin is too low, the uncertainty of the radiation dose delivered to the tumor may increase, leading to poor clinical results. Therefore, for an optimum amount of delivery, image guidance using ExacTrac and CBCT is crucial when treating spinal tumors with SBRT.

Chang et al. [11] reported the setup discrepancies with ExacTrac X-ray 6 degree of freedom (6D) and CBCT on a Novalis Tx system in both phantom and retrospective patient studies. In 16 patients who underwent spinal SBRT, the residual setup error for the root mean square (RMS) between ExacTrac and CBCT was <2.0 mm in patients and <1.0 mm in the phantom for the translational direction, and <1.5˚ in patients and <1.0˚ in the phantom for the rotational direction.

Similarly, in our institution [12], we analyzed the residual setup error between ExacTrac and CBCT in 107 brain tumor patients treated with SRS using Novalis Tx, from August 2012 to July 2016. The difference in the RMS on online matching between 6D ExacTrac and 3D CBCT was 1.01 in the translational direction and 0.82˚ in the rotational direction. However, the limitation of our study was that it only analyzed patients who received intracranial SRS.

Therefore, in this study, we aimed to analyze the difference in the residual setup error between 6D ExacTrac and 3D CBCT image guidance systems in patients with spinal tumors who underwent SBRT.

## Materials and methods

### Study overview

The retrospective data analysis of 76 patients with spinal tumors enrolled in this study was approved by the Institutional Review Board of the Yeungnam University Medical Center (YUMC 2020-09-068). The need for informed consent was waived by the approval of the Institutional Review Board of the Yeungnam University Medical Center (YUMC 2020-09-068), given that patient anonymity was ensured. All methods were carried out in accordance with relevant guidelines and regulations.

### Patient selection

We investigated 76 patients with spinal tumors who received SBRT using the Novalis Tx in our institution between January 2013 and September 2020. The patient and treatment characteristics are described in Table 1. For all 76 patients (female, 40 [53%]; male, 36 [47%]; average age, 62 years) undergoing SBRT, image-guided verification was performed before the radiation treatment using BrainLAB 6D ExacTrac and CBCT. The treatment sites for spinal SBRT were cervical in 11 cases, thoracic in 41 cases, and lumbar in 24 cases. A total of 268 fractions were investigated: 27 Gy in 3 fractions (36 [47%] cases) and 32 Gy in 4 fractions (40 cases).

### Immobilization and CT simulation

For all patients, Vac-lok (EZ-FIX®, Arlico Medical Company, South Korea) fixtures were used to minimize patient movement and to increase interfractional setup reproducibility. Depending on the treatment regions, various assistive devices were used to minimize movement during radiation. For instance, for most patients with tumors in the cervical spine, a head and neck thermal mask (DUON™, Orfit Industries, Wijnegem, Belgium) was used for fixation. CT simulations, with a thickness of 2.5 mm, were performed using a Brilliance Big Bore CT simulator (Philips Inc., Cleveland, OH).

**Table 1. Characteristics of patients and treatment.**

| Patient characteristics | |
| --- | --- |
| Number of patients | N = 76 |
| Median age(Range) | 62(30–82) |
| Gender(%) | |
| Female | 40(53) |
| Male | 36(47) |
| Spinal region of used for the treatment (%) | |
| Cervical | 11(14) |
| Thoracic | 41(54) |
| Lumbar | 24(32) |
| Treatment characteristics | |
| Number of fractions | n = 268 |
| Fraction schemes (dose/fraction) (%) | |
| 27 Gy in 3 fractions (7 Gy) | 36(47) |
| 32 Gy in 4 fractions (8 Gy) | 40(53) |

## Treatment planning and delivery techniques

To increase the sophistication in images for contour delineation, axial T1, axial T2 gadolinium-enhanced T1 of 2 mm thickness, and magnetic resonance image sequences were fused. In the case of spine SBRT with our institution protocol, planning target volume (PTV) was created by extending the margin 3 mm in all directions from the clinical target volume. When major organs such as the spinal cord were adjacent, the PTV was reduced by radiation oncologist. The treatment plan was generated using nine-field static intensity-modulated radiotherapy and anisotropic analytical algorithm. The beam arrangement of the radiation was set in a direction to avoid critical organs, to the best possible extent, with the majority of them directed posteriorly. From April 2019, an upgraded version, Eclipse ARIA 15.6 (Varian Medical System, Palo Alto, CA, USA) instead of Eclipse ARIA 8.6 (Varian Medical System) was used for treatment planning. The utilized radiation treatment plan satisfied the TG-101 guidelines for the normal tissue tolerance dose [13].

## Image registration and setup protocol

Fig 1 shows the positioning for the patient setup for T spinal stereotactic body radiation (SBRT) using BrainLAB Infrared (IR) Reflective Reference Star (BrainLAB, AG, Feldkirchen, Germany). Image registration was performed using BrainLAB's 6D ExacTrac and Varian's 3D CBCT. Using the images, shifts were calculated in the translational (ExacTrac and CBCT: lateral, longitudinal, and vertical) and rotational (ExacTrac: pitch, roll, and yaw direction; CBCT: yaw) directions. Fig 2 indicates image registration using ExacTrac tube 1 and the ExacTrac tube 2 with BrainLAB for thoracic SBRT. Image registration using Varian CBCT for thoracic SBRT is shown in Fig 3.

The setup protocol of spinal SBRT followed in our hospital was implemented. First, ExacTrac images were obtained using ExacTrac tubes 1 and 2, followed by image reconstruction by digitally reconstructed radiography simulator system using CT data. Second, the patient was couch-shifted, using the difference value in image registration between the measured ExacTrac image and reconstructed digitally reconstructed radiography image. Third, it was ensured that the patient had been properly moved using ExacTrac. Image setup tolerance limit was considered to be 0.5 mm for the translational direction and 0.5° for the rotational direction. However, if X-ray verification was performed more than three times and still exceeded the

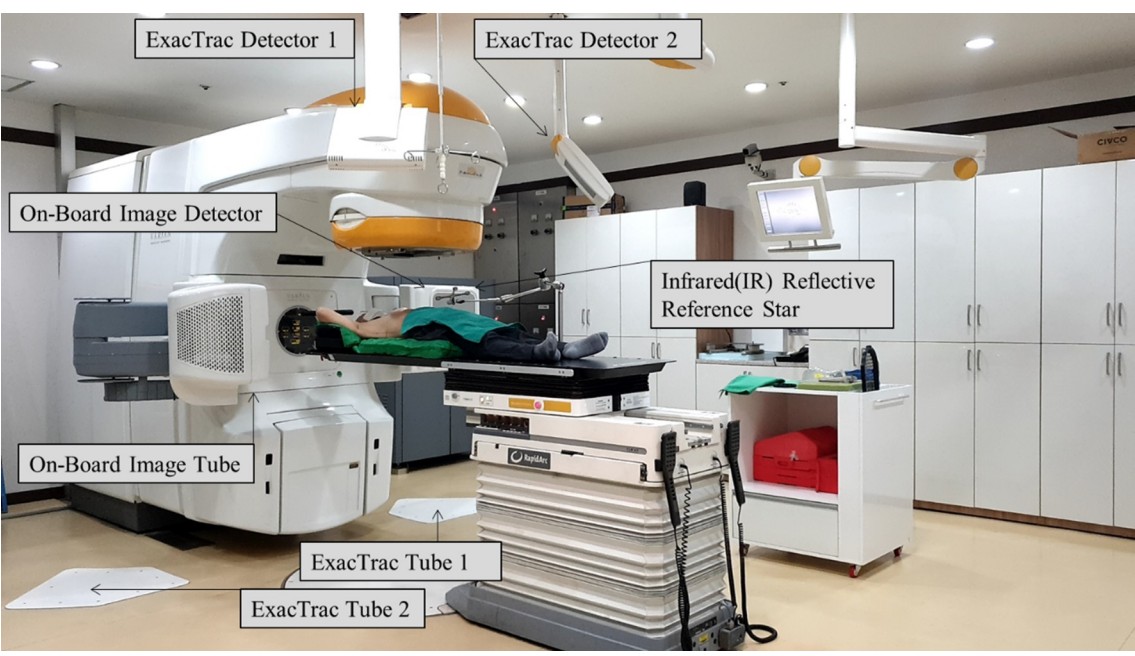

**Fig 1. Positioning for the patient setup for thoracic spinal stereotactic body radiation therapy using BrainLAB Infrared (IR) Reflective Reference Star (BrainLAB, AG, Feldkirchen, Germany).**

tolerance limit, then it was ignored. Fourth, 3D CBCT images were acquired from the patient's exact position from which the ExacTrac images were obtained. Finally, after the radiation oncologist confirmed the patient's setup position, it was finally corrected using the 3D CBCT and radiotherapy was performed.

### Analysis of the residual setup errors between ExacTrac and CBCT

The difference in the residual setup error between ExacTrac and CBCT, for a total of 268 fractionations, was analyzed as RMS, standard deviation, and difference in the translational (lateral, longitudinal, and vertical) and rotational (yaw) directions. In addition, RMS values measured in each image were statistically analyzed through paired t-tests using the SPSS software (IBM Corp., Chicago, IL, USA). A p-value of less than 0.05 was considered statistically significant.

## Results

In the exact patient position, before radiation, the average value obtained by subtracting the 6D ExacTrac value from the 3D CBCT value was -0.75 mm, 0.45 mm, 0.16 mm, and -0.03˚; the RMS value was 1.31 mm, 1.06 mm, 0.87 mm, and 0.64˚; and the standard deviation was 0.80 mm, 0.72 mm, 0.62 mm, and 0.44˚ for lateral, longitudinal, vertical, and yaw directions, respectively. Histograms and normalized curves for the translational and rotational directions between the 6D ExacTrac and 3D CBCT are shown in Fig 4.

Table 2 shows the residual setup errors between 6D ExacTrac and 3D CBCT for spinal SBRT. The present study results were compared with the results from a study by Chang et al. [11] on spinal tumors and those of the brain SRS study performed at our institution [12].

The paired t-test of the RMS values of the residual setup error in 6D ExacTrac and 3D CBCT showed significant differences in the lateral, longitudinal, and vertical directions, but not in the yaw direction.

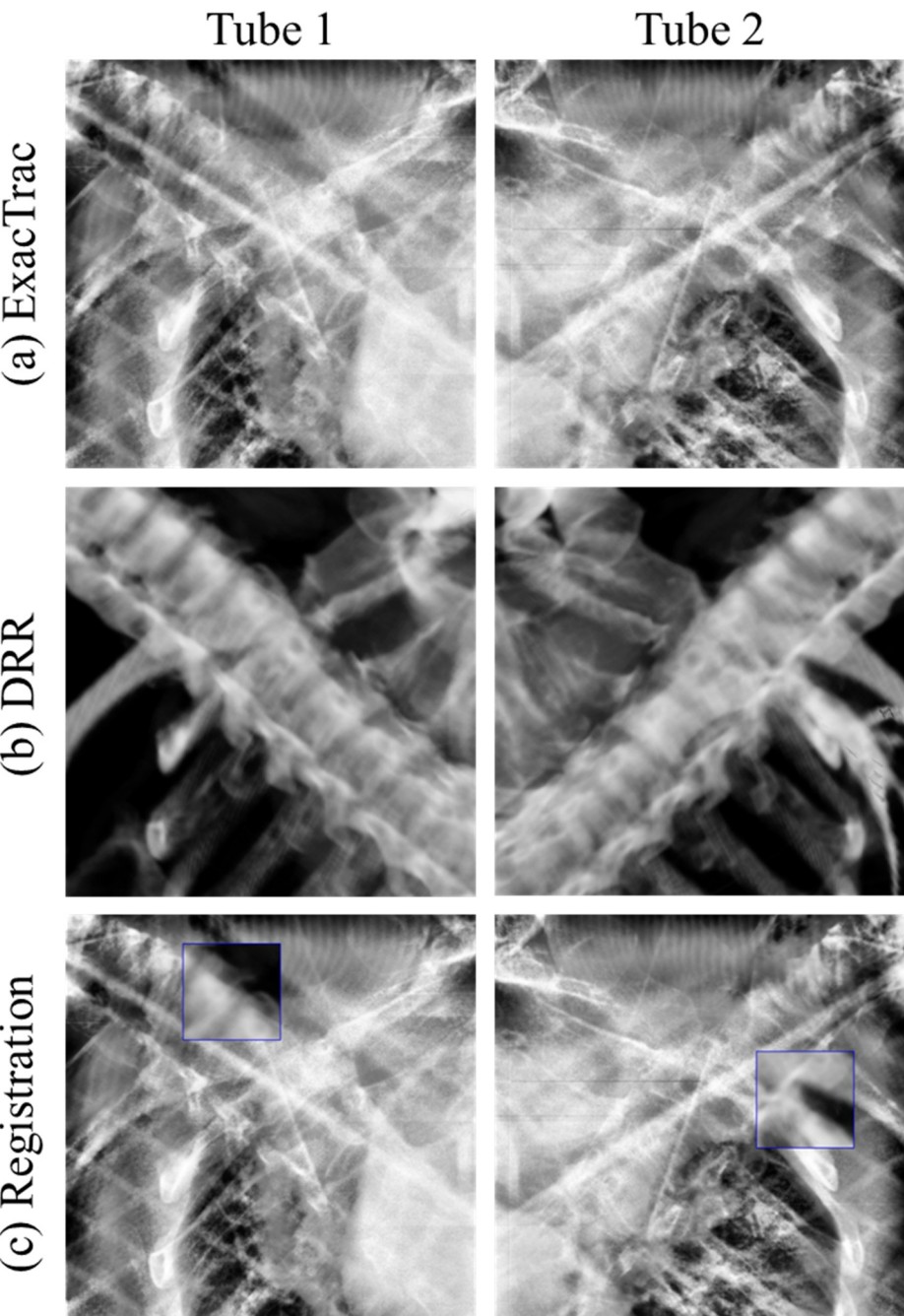

**Fig 2. Image registration using ExacTrac tube 1 and ExacTrac tube 2 with BrainLAB for thoracic spinal stereotactic body radiation therapy.** (a) ExacTrac image; (b) digitally reconstructed radiography image; (c) registration image.

According to the region, the spine was divided into cervical, thoracic, and lumbar regions, and the difference in the residual setup error between 6D ExacTrac and CBCT images was analyzed. The results are shown in S1 Table. In particular, the cervical spine showed a significant difference between ExacTrac and CBCT in all directions (lateral, longitudinal, vertical, and yaw).

## (a)Planning CT  (b)CBCT  (c)Registration

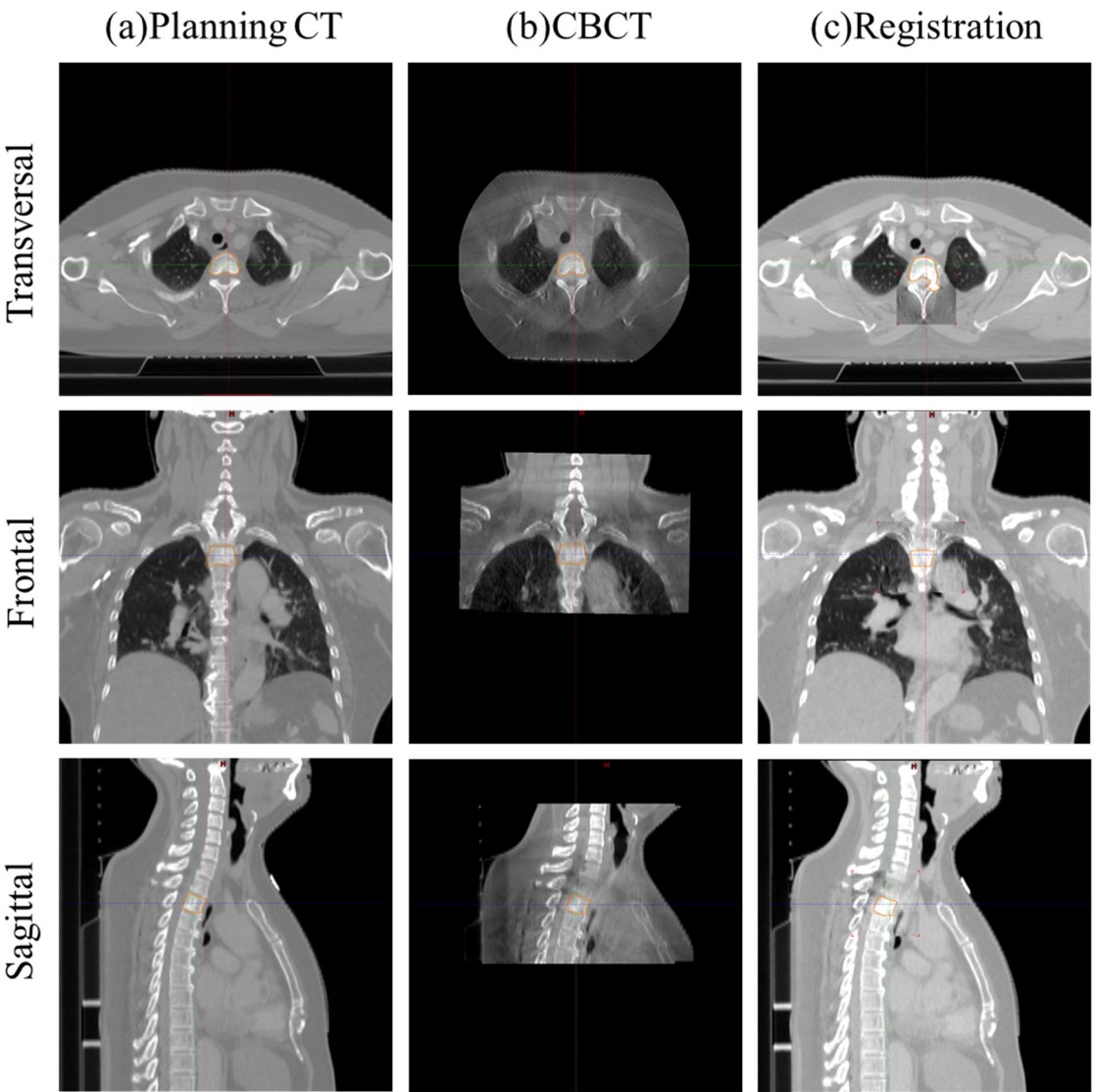

**Fig 3. Image registration using Varian cone-beam computed tomography (CBCT) for thoracic spinal stereotactic body radiation therapy.** (a) planning computed tomography (CT); (b) CBCT; (c) registration image.

## Discussion

We analyzed the images of patients treated with spine SBRT using ExacTrac and CBCT at the exact same location and determined the difference in residual setup errors between ExacTrac and CBCT images. In 76 patients who underwent SBRT, a total of 268 fractions of images were analyzed. The results showed that the difference in average RMS values between ExacTrac and CBCT was <1.03 mm in the translational direction and <0.47° in the rotational direction; the differences were statistically significant in the lateral, longitudinal, and vertical directions but not in the yaw direction. Based on these results, we suggest that ExacTrac image findings should be further confirmed using CBCT.

Recent imaging-related technologies have garnered a lot of attention recently; they have been used to improve the accuracy of radiotherapy. Particularly, the use of ExacTrac and CBCT has become widespread. Previously, various studies have evaluated the difference between ExacTrac and CBCT in phantom, intracranial, and spinal SRS.

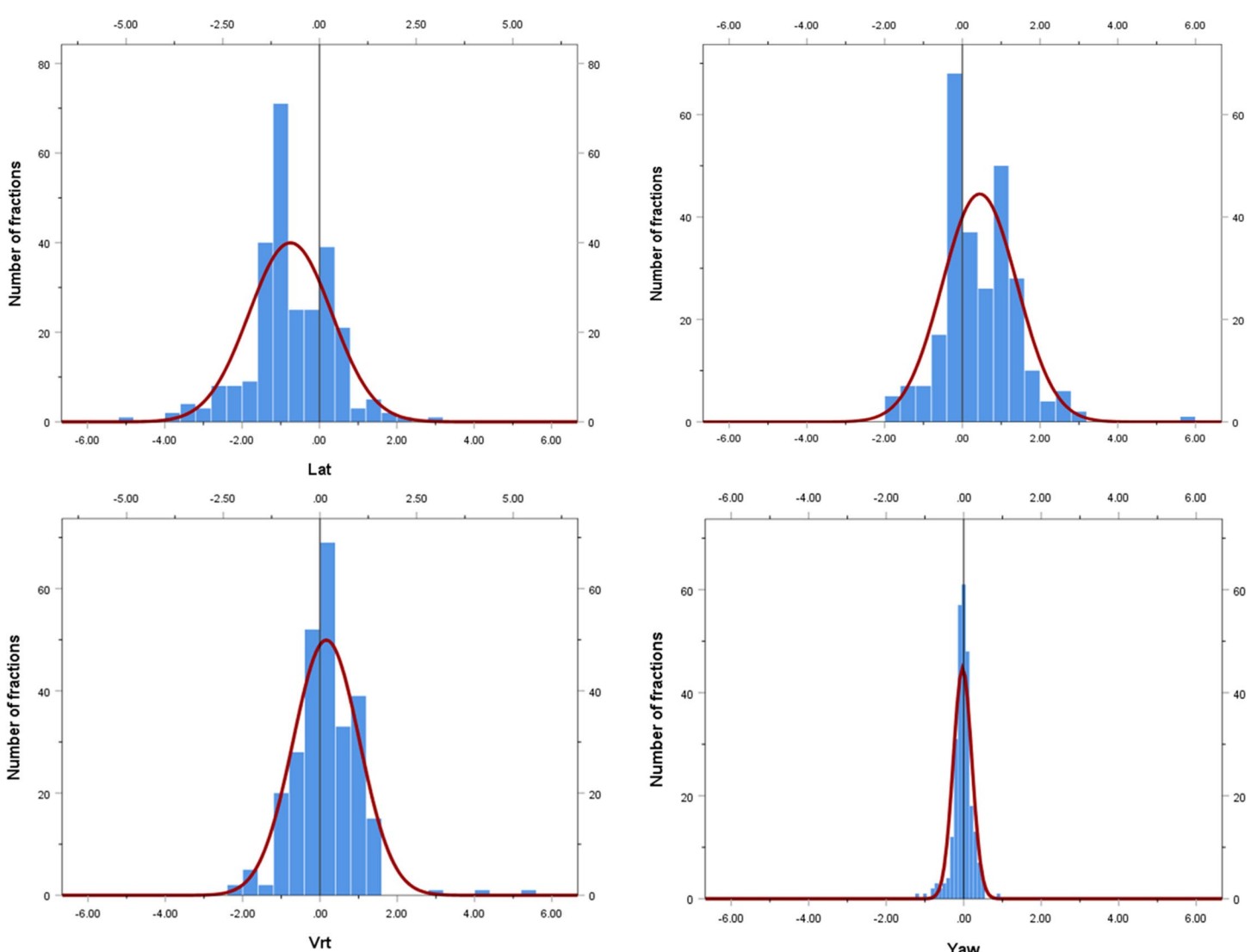

**Fig 4. Histogram and normalized curves for the translational and rotational directions between 6D ExacTrac and 3D cone-beam computed tomography.** (a) lateral; (b) longitudinal; (c) vertical; and (d) yaw directions.

The difference between the ExacTrac and CBCT images of the spine was studied by Chang et al. [11]. They evaluated 16 cases of spinal SBRT in 11 patients, and the difference in the RMS was found to be <2.0 mm in the translation direction and <1.5˚ in the rotational direction. They analyzed the cases of phantoms and patients separately. In the case of phantoms, there were significant differences in the ExacTrac and CBCT images in the vertical, lateral, and yaw directions, but not in the longitudinal, pitch, and roll directions. However, in the case of patients, there were no significant differences in RMS values between ExacTrac and CBCT in all directions (vertical, longitudinal, lateral, pitch, roll, and yaw). The authors believed that the differences between ExacTrac and CBCT images in spine SBRT, though small, were of clinical significance. Of note, their study's major drawback was the limited number of cases. In addition, Chang et al. suggested that the following are possible sources of residual setup discrepancy between ExacTrac and CBCT: (1) inter-scan patient motion, (2) the difference in image

**Table 2. Residual setup errors between 6D ExacTrac and 3D CBCT for spinal SBRT.**

| | Region | Number of patient | Directions | Setup error for 6D ExacTrac | | Setup error for 3D CBCT | | 6D ExacTrac vs 3D CBCT | | |
| --- | --- | --- | --- | --- | --- | --- | --- | --- | --- | --- |
| | | | | | | | | Difference | | p-value of paired t-test |
| | | | | RMS | SD | RMS | SD | RMS | SD | |
| **Chang et al. [11]** | Spine | N = 11, n = 16 | Translational | | | | | | | |
| | | | Lateral (x-axis) (mm) | n/a | n/a | n/a | n/a | 1.22 | 1.25 | 0.875 |
| | | | Longitudinal (z-axis) (mm) | n/a | n/a | n/a | n/a | 0.93 | 0.94 | 0.307 |
| | | | Vertical-(y-axis)(mm) | n/a | n/a | n/a | n/a | 1.67 | 1.65 | 0.162 |
| | | | Rotational | | | | | | | |
| | | | Pitch (x-axis) (˚) | n/a | n/a | n/a | n/a | 0.54 | 0.53 | 0.271 |
| | | | Roll (z-axis) (˚) | n/a | n/a | n/a | n/a | 0.40 | 0.40 | 0.356 |
| | | | Yaw (y-axis) (˚) | n/a | n/a | n/a | n/a | 0.87 | 0.90 | 0.896 |
| **Oh et al. [12]** | Brain | N = 107, n = 138 | Translational | | | | | | | |
| | | | Lateral (x-axis) (mm) | 0.20 | 0.20 | 0.97 | 0.65 | 1.01 | 0.60 | <0.001[a] |
| | | | Longitudinal (z-axis) (mm) | 0.24 | 0.24 | 0.77 | 0.77 | 0.84 | 0.63 | 0.425 |
| | | | Vertical (y-axis) (mm) | 0.20 | 0.20 | 0.76 | 0.75 | 0.76 | 0.56 | 0.028[a] |
| | | | Rotational | | | | | | | |
| | | | Pitch (x-axis) (˚) | 0.18 | 0.18 | n/a | n/a | n/a | n/a | n/a |
| | | | Roll (z-axis) (˚) | 0.17 | 0.17 | n/a | n/a | n/a | n/a | n/a |
| | | | Yaw (y-axis) (˚) | 0.22 | 0.22 | 0.80 | 0.80 | 0.82 | 0.49 | 0.226 |
| **Present Study** | Spine | N = 76, n = 268 | Translational | | | | | | | |
| | | | Lateral (x-axis) (mm) | 0.40 | 0.40 | 1.10 | 0.81 | 1.31 | 0.80 | <0.001 |
| | | | Longitudinal (z-axis) (mm) | 0.36 | 0.36 | 0.89 | 0.77 | 1.06 | 0.72 | <0.001 |
| | | | Vertical (y-axis) (mm) | 0.33 | 0.33 | 0.95 | 0.94 | 0.87 | 0.62 | 0.002 |
| | | | Rotational | | | | | | | |
| | | | Pitch (x-axis) (˚) | n/a | n/a | n/a | n/a | n/a | n/a | n/a |
| | | | Roll (z-axis) (˚) | n/a | n/a | n/a | n/a | n/a | n/a | n/a |
| | | | Yaw (y-axis) (˚) | 0.30 | 0.30 | 0.46 | 0.46 | 0.64 | 0.44 | 0.526 |

Residual setup errors between the 6D ExacTrac and 3D cone-beam computed tomography (CBCT) in spinal stereotactic body radiation therapy (SBRT). [a]$p < 0.05$. N, number of patients; n, number of fractions; RMS, root mean square; SD, standard deviation.

fusion algorithm, (3) poorer visibility of anatomical structures in digitally reconstructed radiograph and X-ray images, and (4) discrepancy of geometric accuracy for the two systems.

Before this study, we analyzed intracranial SRS with 138 fractions in 107 patients [12]. A significant difference was noted only in the vertical and lateral directions, whereas no significant difference was observed in the longitudinal and yaw directions. However, the previous study was restricted to intracranial sites. Therefore, in this study, we investigated a higher number of cases of spinal SBRT, in contrast to the reduced number of cases reported by Chang et al. [11] and non-inclusion of extracranial sites in our previous study.

Our study analyzed residual setup errors according to the region of the spine involved, something that has not been studied previously; the spine was divided into cervical, thoracic, and lumbar regions to analyze the residual setup error between 6D ExacTrac and 3D CBCT. Notably, in the cervical spine region, there was a significant difference between ExacTrac and CBCT in all directions (lateral, longitudinal, vertical, and yaw). Therefore, particularly in the cervical spine, it is necessary to verify ExacTrac images using CBCT data.

This study has a few limitations. There may be patient movement between ExacTrac and CBCT, thus affecting the findings. In brain SRS, a frameless mask was used to immobilize the patient, whereas in spine SBRT, head and neck masks or Vac-lok were used, depending on the

treatment site. The possibility of movement may be greater with the frameless masks for SRS than with the head and neck masks for SBRT. Moreover, this study analyzed the difference between the residual setup errors of 6D ExacTrac and 3D CBCT only through online review. In an offline review of 6D CBCT, it would have been possible to analyze 6D data (lateral, longitudinal, and vertical in the translational direction; and pitch, roll, and yaw in the rotational direction). However, as unwanted records were recorded on the server in offline review, they were excluded from this study.

## Conclusions

Taken together, this study analyzed images of 268 fractions in 76 patients who received spine SBRT. Significant differences in the RMS were observed in the lateral, longitudinal, and vertical directions between ExacTrac and CBCT, whereas no significant difference was noted in the yaw direction. Therefore, for spine radiotherapy in clinical practice, verification with CBCT after ExacTrac image acquisition is essential.

## Supporting information

**S1 Table.**
(DOCX)

## Author Contributions

**Conceptualization:** Jaehyeon Park, Ji Woon Yea, Jae Won Park, Se An Oh.

**Data curation:** Jaehyeon Park.

**Formal analysis:** Jaehyeon Park.

**Funding acquisition:** Se An Oh.

**Investigation:** Jaehyeon Park.

**Methodology:** Ji Woon Yea, Jae Won Park.

**Project administration:** Se An Oh.

**Resources:** Se An Oh.

**Supervision:** Se An Oh.

**Validation:** Jaehyeon Park, Ji Woon Yea, Jae Won Park, Se An Oh.

**Writing – original draft:** Jaehyeon Park, Se An Oh.

**Writing – review & editing:** Jaehyeon Park, Ji Woon Yea, Jae Won Park, Se An Oh.

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
