## [Decision Letter · Decision Letter 0]

29 Apr 2021

PONE-D-21-03969

Evaluation of the setup discrepancy between 6D ExacTrac and cone beam computed tomography in spine stereotactic body radiation therapy

PLOS ONE

Dear Dr. Oh,

Thank you for submitting your manuscript to PLOS ONE. After careful consideration, we feel that it has merit but does not fully meet PLOS ONE’s publication criteria as it currently stands. Therefore, we invite you to submit a revised version of the manuscript that addresses the points raised during the review process.

We look forward to receiving your revised manuscript.

Kind regards,

Stephen Chun

Academic Editor

PLOS ONE

Journal Requirements:

2. . Please include captions for your Supporting Information files at the end of your manuscript, and update any in-text citations to match accordingly. Please see our Supporting Information guidelines for more information: http://journals.plos.org/plosone/s/supporting-information.

Reviewers' comments:

Reviewer's Responses to Questions

**Comments to the Author**

1. Is the manuscript technically sound, and do the data support the conclusions?

Reviewer #1: Yes

Reviewer #2: Yes

2. Has the statistical analysis been performed appropriately and rigorously? 

Reviewer #1: Yes

Reviewer #2: Yes

3. Have the authors made all data underlying the findings in their manuscript fully available?

Reviewer #1: Yes

Reviewer #2: Yes

4. Is the manuscript presented in an intelligible fashion and written in standard English?

Reviewer #1: Yes

Reviewer #2: Yes

5. Review Comments to the Author

Reviewer #1: Although there was a previous study addressing a similar question, it was of much smaller scale and there was no region-specific (C-spine vsT-spine vs L-spine) data. Therefore, this study will provide very important info that will guide spine SBRT practice.

1.Vac-lok fixtures refers to just a cradle and not BodyFix, correct? From the very small picture, it looks like it was just a regular Vac-lok. Please clarify.

2. I presume that an Exac-Trac registration was done followed by a CBCT registration and the final shifts were based on CBCT. Please clarify.

3. Table 2- Please fix spacing of the table.

Reviewer #2: Manuscript is sound and presents valuable data that is very relevant to this widespread technology. It may be beneficial to include what margins were used for PTV expansion and whether the magnitude of uncertainty in the setup error would have any clinically significant effects on the dosimetry of the spine SBRT plan (PTV coverage, dose to OAR's, etc). Additionally, what factors are contributing to the residual setup discrepancy? Potential patient movement between ExacTrac and CBCT was mentioned...are there any other potential reasons (insufficient immobilization, anatomical motion, differences in image quality, etc)? Do these factors differ between cervical, thoracic and lumbar spine regions?

6. PLOS authors have the option to publish the peer review history of their article (what does this mean?). If published, this will include your full peer review and any attached files.

Reviewer #1: No

---

## [Author Response · Author response to Decision Letter 0]

7 May 2021

May 2021

Editorial Board

PLoS One

Dear Editor:

We would like to re-submit the attached manuscript entitled “Evaluation of the setup discrepancy between BrainLAB 6D ExacTrac and cone-beam computed tomography on the image guide system of the Novalis-Tx for spine stereotactic body radiation therapy.” The manuscript ID is PONE-D-21-03969.

The manuscript has been carefully rechecked and appropriate changes (red text in the revised manuscript) have been made in accordance with the reviewers’ suggestions. The responses to their comments have been prepared and attached herewith. 

We thank you and the reviewers for the thoughtful suggestions and insights, which have enriched the manuscript and produced a more balanced and better account of the research. We hope that the revised manuscript is now suitable for publication in your journal.

I look forward to your reply.

Sincerely,

Se An Oh

Department of Radiation Oncology

Yeungnam University College of Medicine

Daegu, South Korea 

sean.oh5235@gmail.com

---

## [Editor Report · Decision Letter 1]

12 May 2021

Evaluation of the setup discrepancy between 6D ExacTrac and cone beam computed tomography in spine stereotactic body radiation therapy

PONE-D-21-03969R1

Dear Dr. Oh,

We’re pleased to inform you that your manuscript has been judged scientifically suitable for publication and will be formally accepted for publication once it meets all outstanding technical requirements.

Kind regards,

Stephen Chun

Academic Editor

PLOS ONE
---

## [Editor Report · Acceptance letter]

18 May 2021

PONE-D-21-03969R1 

Evaluation of the setup discrepancy between 6D ExacTrac and cone beam computed tomography in spine stereotactic body radiation therapy 

Dear Dr. Oh:

I'm pleased to inform you that your manuscript has been deemed suitable for publication in PLOS ONE. Congratulations! Your manuscript is now with our production department. 

Kind regards, 

on behalf of

Dr. Stephen Chun 

Academic Editor

PLOS ONE